# Igf2bp3 maintains maternal RNA stability and ensures early embryo development in zebrafish

Fan Ren [1,4], Qiaohong Lin [1,4], Gaorui Gong [1], Xian Du [2], Hong Dan [1], Wenying Qin [2], Ran Miao [1], Yang Xiong [1], Rui Xiao[2], Xiaohui Li [1], Jian-Fang Gui [1,3] & Jie Mei [1✉]

Early embryogenesis relies on maternally inherited mRNAs. Although the mechanism of maternal mRNA degradation during maternal-to-zygotic transition (MZT) has been extensively studied in vertebrates, how the embryos maintain maternal mRNA stability remains unclear. Here, we identify Igf2bp3 as an important regulator of maternal mRNA stability in zebrafish. Depletion of maternal *igf2bp3* destabilizes maternal mRNAs prior to MZT and leads to severe developmental defects, including abnormal cytoskeleton organization and cell division. However, the process of oogenesis and the expression levels of maternal mRNAs in unfertilized eggs are normal in maternal *igf2bp3* mutants. Gene ontology analysis revealed that these functions are largely mediated by Igf2bp3-bound mRNAs. Indeed, Igf2bp3 depletion destabilizes while its overexpression enhances its targeting maternal mRNAs. Interestingly, *igf2bp3* overexpression in wild-type embryos also causes a developmental delay. Altogether, these findings highlight an important function of Igf2bp3 in controlling early zebrafish embryogenesis by binding and regulating the stability of maternal mRNAs.

[1] College of Fisheries, Huazhong Agricultural University, 430070 Wuhan, China. [2] Frontier Science Center for Immunology and Metabolism, Medical Research Institute, and Department of Hematology, Zhongnan Hospital of Wuhan University, Wuhan University, 430071 Wuhan, China. [3] State Key Laboratory of Freshwater Ecology and Biotechnology, Institute of Hydrobiology, Chinese Academy of Sciences, University of the Chinese Academy of Sciences, 430072 Wuhan, China. [4] These authors contributed equally: Fan Ren, Qiaohong Lin. ✉email: jmei@mail.hzau.edu.cn

Maternally inherited mRNAs direct early developmental events in vertebrates, such as egg activation, fertilization, and embryonic cell division, along with the assembly of cortical cytoskeleton[1–4]. The post-transcriptional regulatory mechanisms that control maternal mRNA stability and degradation are fine-tuning and complex. In recent years, the degradation of maternal mRNA during the stage of maternal-to-zygotic transition (MZT) has been extensively studied in vertebrates, and it was found that some factors such as RNA-binding proteins (RBPs), microRNA, or codon usage mediate maternal RNA degradation[5–8]. How maternal mRNAs maintain stability prior to MZT (pre-MZT) is one of the long-standing questions in the field of developmental and reproductive biology.

mRNA stability is usually modulated by their 5′ cap and 3′ poly (A) tail length[9,10]. Recently, N6-methyladenosine (m6A), the most prevalent modification in eukaryotic mRNA, has been correlated with mRNA stability. The fate of m6A-modified mRNA was mostly determined by the "reader" proteins that were potential m6A-binding protein[11]. Based on their mechanisms for selective binding to m6A-modified RNAs, the m6A reader proteins are divided into three classes, such as the m6A reader proteins containing a YTH domain (class I), heterogeneous nuclear ribonucleoproteins (hnRNPC and hnRNPG, class II), and Insulin-like growth factor 2 mRNA-binding proteins (IGF2BP1–3, class III)[11]. The m6A reader YTHDF2 (YTH-domain family member 2) selectively recognized m6A-containing mRNAs and caused their degradation[12], whereas IGF2BPs enhanced the stability and translation of m6A-containing mRNAs[13]. HnRNPs selectively bound to m6A-containing RNAs through an m6A-switch mechanism and mediated RNA splicing[14,15]. YTHDF2 has been shown to regulate maternal transcriptome during oocyte maturation and establish oocyte competent to ensure early embryo development in mice[16].

In the early zebrafish embryo, maternal mRNAs are degraded by a ythdf2-mediated pathway during the zebrafish MZT[6]. The IGF2BPs have been reported as an RNA stabilizer in the P-body to enhance the stability of mRNAs in human cells[11,13]. However, the functions of IGF2BPs are not clear in vivo, since there is no report about gene knockout studies of IGF2BPs in vertebrates. Therefore, we generated igf2bp1–3 mutant zebrafish employing CRISPR/Cas9 technology and found that only igf2bp3 mutant zebrafish displayed developmental defects during early embryogenesis. We observed that maternal-effect igf2bp3 mutant (Migf2bp3) embryos had severe defects during early embryo development and displayed a significant reduction of maternal mRNA levels at pre-MZT. Our results indicate that Igf2bp3 binds to maternal mRNAs and maintains their stability, thus adding to our knowledge about the regulatory mechanism for the fate of maternal mRNA during early embryo development.

## Results

**Maternal igf2bp3 is essential for early embryo development.** Tissue distribution of igf2bp3 transcript was detected by qRT-PCR. The zebrafish igf2bp3 was predominantly expressed in the ovary (Supplementary Fig. 1). To investigate the function of igf2bp3 during zebrafish development, we generated two igf2bp3 mutant lines with 2 bp deletion (named igf2bp3Δ2) and 41 bp deletion (named igf2bp3Δ41) using CRISPR-Cas9 technology (Fig. 1a). The genome sequencing results of F2 generation was shown to confirm the DNA lesion in the igf2bp3 mutants (Supplementary Fig. 2). Zebrafish Igf2bp3 protein consisted of six functional domains including two RRM domains and four KH domains. The mutations in igf2bp3Δ2 and igf2bp3Δ41 resulted in the open reading frame (ORF) shift and caused premature termination codons (Fig. 1b).

Homozygotic igf2bp3 mutants were morphologically normal, viable, and fertile. However, most maternal homozygous mutants of igf2bp3Δ2 and igf2bp3Δ41 lines displayed severe defects during early embryo development. In the following studies, we named the maternal igf2bp3 mutants as Migf2bp3. In wild type, the yolk granules coalesced during egg activation and the blastoderm was transparent at 2- and 4-cell stages, whereas the mutants displayed a defect in yolk coalescence and enriched opaque grains in the blastoderm (Fig. 1c). Early cleavage stages were characterized by the invagination of the adhesive membrane with the formation of the cleavage furrow. In Migf2bp3 embryos, cleavage furrow ingression appeared normal before 8-cell stage. A clearly visible membrane septum was observed at 8-cell stage in wild-type embryos (black arrow), whereas the Migf2bp3 embryos lacked clearly defined septum (white arrow). Compared with a cellularized blastula in wild-type embryos at 32- and 512-cell stage, Migf2bp3 embryos exhibited non-adhesive blastomeres with rounded morphology (red arrow) (Fig. 1c). Statistically, both the igf2bp3Δ2 and igf2bp3Δ41 mutants displayed the same phenotype of defective cell division at 512-cell stage (Fig. 1d). At 1k-cell stage and 26 hpf, the Migf2bp3 embryos exhibited various degrees of defects and were classified according to the extent of cellularization in the embryo (Supplementary Fig. 3). Whole-mount in situ hybridization (WISH) was conducted to examine the expression of igf2bp3 mRNA in Migf2bp3 embryos. As shown in Supplementary Fig. 4, the expression signals of igf2bp3 mRNA dramatically decreased compared with wild-type zebrafish. igf2bp3Δ2 mutants were used in the following studies.

**Migf2bp3 embryos display defects in cytoskeleton assembly.** During early embryo development, defective cell division was observed in the Migf2bp3 embryos. F-actin, a composition of the contractile ring apparatus, was normally recruited to the cleavage furrow and essential for the generation of adhesive cell walls[17,18]. During the first embryonic division in wild-type zebrafish embryos, F-actin was recruited to and accumulated along the furrow to form the contractile ring (Fig. 2a). However, the amount of F-actin in the contractile ring was greatly reduced in Migf2bp3 embryos during furrow initiation. During cytokinesis at 4- and 8-cell stages, F-actin was concentrated at the cleavage furrow to form an adhesive cell wall in wild-type zebrafish embryos. The Migf2bp3 embryos exhibited decreased accumulation of F-actin in the cleavage furrow. Moreover, cytokinesis failure with lack of blastomere coherence appeared by the third cell cycle (Fig. 2a).

Next, we checked the tubulin structure in the embryos (Supplementary Fig. 5). During the initiation of the cleavage furrow, the furrow microtubule arrays (FMA) were arranged parallel to each other and perpendicular to the furrow in both wild type and Migf2bp3 embryos (Supplementary Fig. 5a, i). As the furrow matured, the microtubule gradually enriched at the distal end of furrows with tilted angles to form V-shaped structures (Supplementary Fig. 5b, j). A similar process was also shown in the second cleavage stage (Supplementary Fig. 5c, d, k, l). The result indicated that Migf2bp3 embryos displayed a normal microtubule dynamic. Meanwhile, we investigated whether igf2bp3 depletion affects DNA structure and found that Migf2bp3 embryos displayed regular nuclear division as the wild type at 4-cell stage (Supplementary Fig. 6).

β-Catenin, a main component of cell adhesion junctions, can be labeled to visualize the formation of mature adhesive cell wall. In wild-type embryos at 2-cell stage, β-catenin aggregated and localized along the cleavage plane in mature furrows. In contrast, distribution of the cleavage furrow associated β-catenin was disrupted in Migf2bp3 embryos (Fig. 2b). At 4- and 8-cell stages,

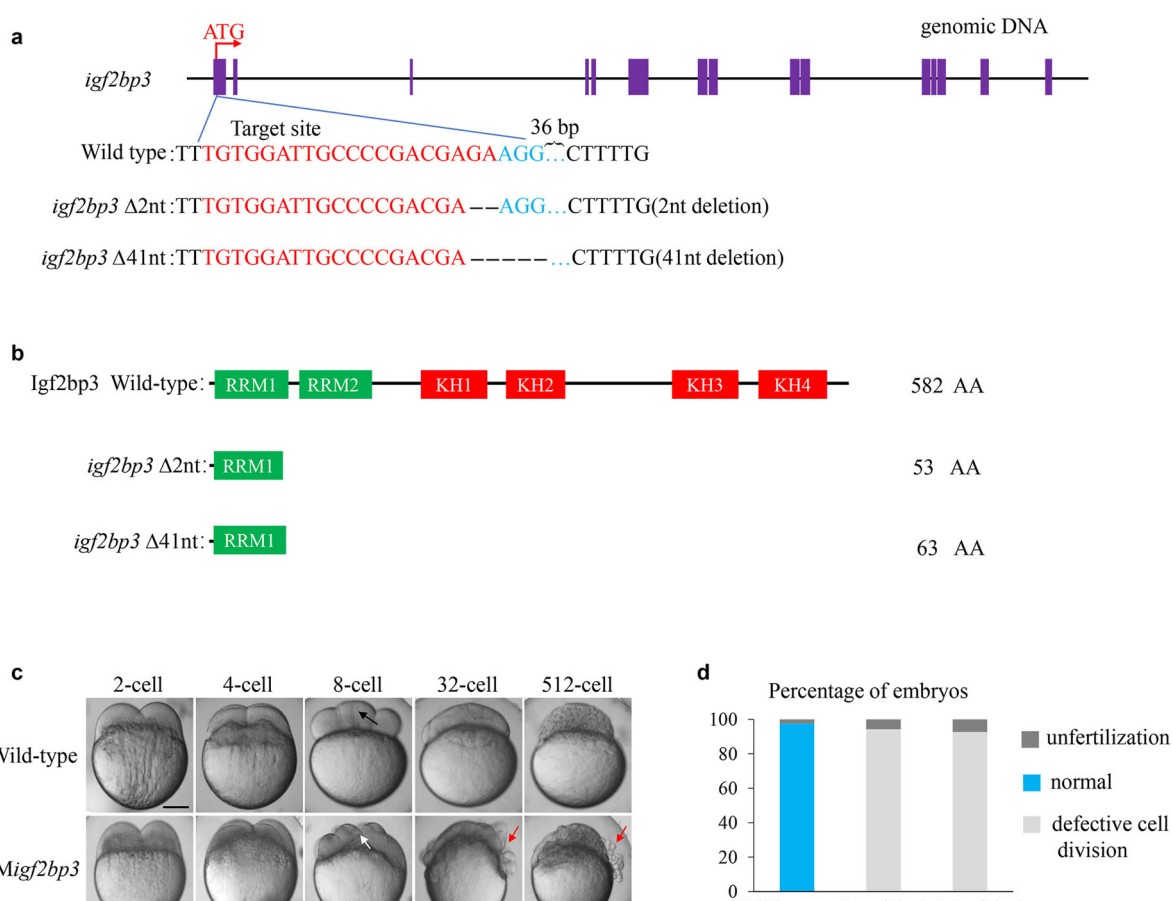

**Fig. 1 Generation and characterization of *igf2bp3* mutant zebrafish. a** Design and mutation types of *igf2bp3* based on CRISPR/Cas9 biotechnology. Exons were represented by purple boxes and the sgRNA target sequence was shown in red. Two types of mutants were generated. **b** Illustration of deduced protein structure of wild-type *igf2bp3* and two mutated *igf2bp3*. **c** The developmental timecourse of wild type and M*igf2bp3* embryos. M*igf2bp3* embryos displayed an inapparent septum (white arrow) compared to that in wild type (black arrow) at 8-cell stage. The non-adhesive cells in the mutants were indicated by red arrows at 32- and 512-cell stage. **d** Statistical analysis of the phenotypes of M*igf2bp3* and wild-type zebrafish embryos at 512-cell stage, as indicated in **c**. Wild type, $N = 5$, $n = 708$; M*igf2bp3* (Δ2nt), $N = 6$, $n = 672$; M*igf2bp3* (Δ41nt), $N = 8$, $n = 749$ ($N$ = female individuals, $n$ = embryos). Scale bars: 200 μm.

we observed abnormal cell division with disrupted and asymmetric furrow ingression in M*igf2bp3* embryos. Moreover, the expression of β-catenin was severely reduced in the furrows of M*igf2bp3* embryos.

Sperm–egg fusion triggers the process of cortical granule cortical granule exocytosis (CGE) depending on the reorganization of F-actin network in which CGs are embedded[19,20]. Since the organization of F-actin was disrupted in the cleavage furrow, the distribution and exocytosis of CGs were quantified by labeling CGs with TRITC-conjugated *Maclura pomifera* agglutinin (MPA), which could determine the size and density of CG vesicles[4]. At 2 min post fertilization (mpf), the size and density of CGs were similar between the wild type and M*igf2bp3* embryos (Fig. 2c). However, the exocytosis process was blocked at later stages in M*igf2bp3* embryos. At 10 mpf, a large number of unreleased CGs was retained in M*igf2bp3* embryos, instead of almost completed CGs exocytosis in wild-type embryo. These results indicated that *igf2bp3* regulated CGs release during early embryo development.

**Accelerated decay of maternal RNAs in M*igf2bp3* embryos.** To obtain the global profile of mRNA expression during early zebrafish embryogenesis, we performed RNA sequencing (RNA-seq)

on wild type and M*igf2bp3* embryos at six different developmental stages including unfertilized egg, 1-cell, 4-cell, 64-cell, sphere and shield stages. According to previous studies, transcripts could be categorized into three superclusters including maternal, semi-stable and zygotic[6]. We found that the RNA abundance changes of maternal mRNAs (from unfertilized egg to 1-cell and 1-cell to 4-cell) in the M*igf2bp3* embryos were markedly greater than those in the wild-type embryos, suggesting that loss of maternal *igf2bp3* function led to accelerated mRNA decay (Fig. 3a, b). In total, 2282 and 2574 maternal genes were identified to be downregulated in M*igf2bp3* embryos compared to control embryos at 1- and 4-cell stage, respectively (Fig. 3c, d).

**Oogenesis is largely normal in the *igf2bp3* mutants.** Since obvious defects were detected in M*igf2bp3* eggs after fertilization, we investigated whether *igf2bp3* mutation affects oogenesis by histological analysis. The results of hematoxylin and eosin staining indicated that the morphology and distribution of CG appeared normal in *igf2bp3* mutant oocytes compared to wild type (Fig. 4a, b). Only a single Balbiani body was detected in the stage I oocytes in both wild type and mutants, which indicated that *igf2bp3* mutant oocytes were polarized (Fig. 4c, d). Simultaneously, we observed grossly normal ovaries in *igf2bp3* mutants

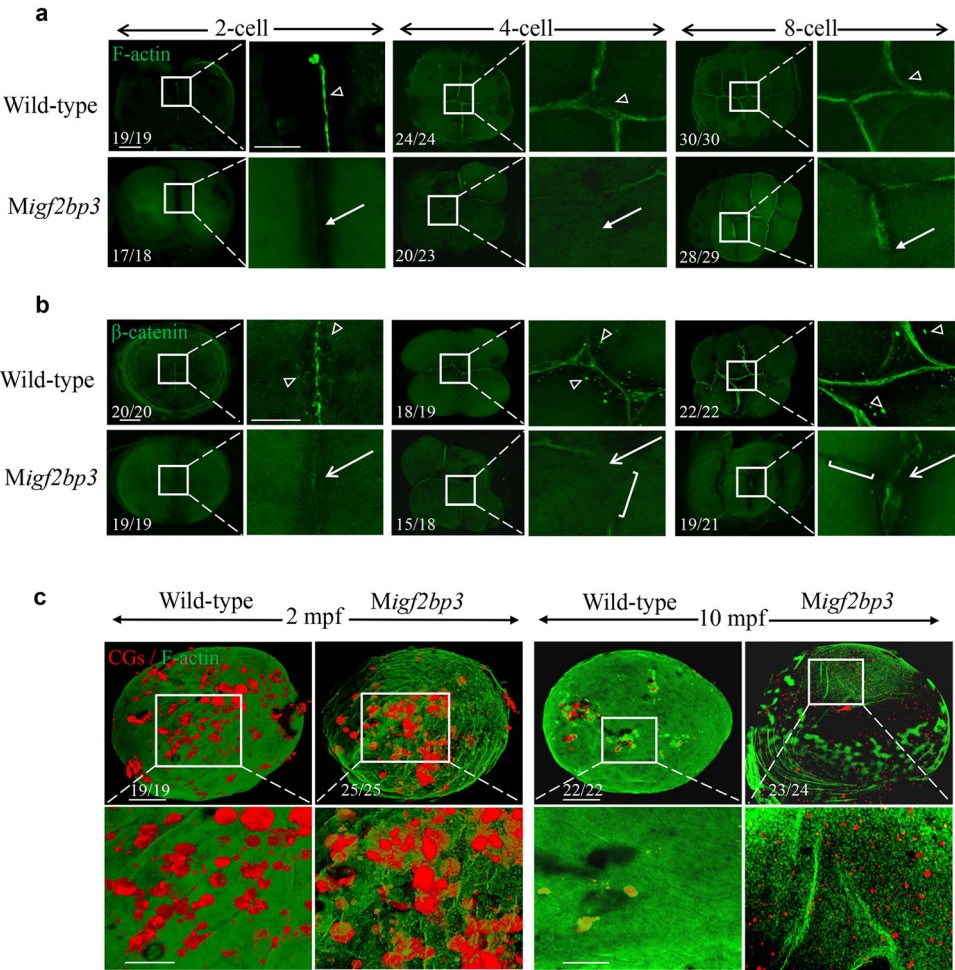

**Fig. 2 M*igf2bp3* embryos exhibit abnormal cytoskeleton organization. a**, **b** Expression and distribution of F-actin and β-catenin in wild type and M*igf2bp3* embryos at 2-, 4-, and 8-cell stages. The F-actin and β-catenin were enriched in mature and apparent cleavage furrow (triangle). The inset showed a zoom-in of the boxed region at each stage. Zoomed-out scale bar = 100 μm; zoomed-in scale bar = 50 μm. **a** M*igf2bp3* embryos at 4-cell stage did neither exhibit intact adhesion junction of F-actin cables nor form the mature furrow (white arrow). **b** M*igf2bp3* embryos exhibited reduced expression of β-catenin (white arrows) and did not form intact cell adhesion junction with defective furrow (white bracket). **c** Double localization of F-actin (green fluorescence) and CG (red fluorescence). The inset showed a zoom-in of the boxed region at each stage. Zoomed-out scale bar = 100 μm; zoomed-in scale bar = 50 μm.

compared with the wild type, indicated by normal morphology of vitelline envelope (VE), CG distribution, structure of the two layers of somatic follicle cells surrounding stage III oocytes of wild type and *igf2bp3* mutants (Fig. 4e, f). Besides, in the unfertilized eggs, there was no significant difference in the expression levels of maternal mRNAs between wild type and *igf2bp3* mutant (Fig. 4g). Furthermore, F-actin content and distribution were the same in the ovary and oocytes of wild type and *igf2bp3* mutants (Fig. 4h, s). In summary, these data proved that the depletion of *igf2bp3* did not affect ovarian and oocyte development.

**Igf2bp3 binds to maternal mRNAs and regulates their stability**. RNA-binding protein immunoprecipitation (RIP) followed by sequencing (RIP-Seq) was performed to identify mRNAs bounded by Igf2bp3, and 3608 target genes were ultimately identified (Supplementary Data 1). Gene ontology (GO) analysis indicated that most of the Igf2bp3 targets were enriched in RNA regulation and metabolism, cell division and cytoskeleton organization, and epigenetic modification processes (Fig. 5a). *Aurkb* (aurora B) has been reported to mediate the process of cytokinesis and cytoskeleton in zebrafish[21]. As a target of Igf2bp3, *aurkb* displayed significant reduction of mRNA expression in M*igf2bp3* embryos, compared to the wild type (Fig. 5b). Moreover, among the target

maternal mRNAs of Igf2bp3, the RNA abundance changes (from unfertilized egg to 1-cell, 1-cell to 4-cell) in the M*igf2bp3* embryos were markedly greater than those in the wild-type embryos, suggesting that loss of maternal *igf2bp3* function led to accelerated mRNA decay of its target maternal mRNAs (Fig. 5c, d). These results suggested that Igf2bp3 binds to the maternal RNAs and regulates their stability, while loss of Igf2bp3 function leads to rapid degradation of maternal mRNAs.

Ythdf2 and zygotically transcribed microRNA miR-430 have been reported to participate in the clearance of maternal mRNAs during zebrafish MZT[5,6,22], while Igf2bp3 maintained the stability of maternal mRNAs during early embryogenesis. To investigate the mRNA degradation mechanism in M*igf2bp3* embryos, comprehensive analysis of RNA-seq data was carried out to reveal the mRNA degradation profile of M*igf2bp3* embryos and miR-430- or Ythdf2-mediated degradation profile[5,6]. As a result, a large majority of the regulatory mRNAs of Igf2bp3, miR-430, and Ythdf2 were different (Supplementary Fig. 7a). Moreover, we compared the target mRNAs of Igf2bp3, Ythdf2, and miR-430, and found that the majority of target mRNAs of each were not overlapped (Supplementary Fig. 7b), suggesting that the accelerated degradation of maternal mRNAs in M*igf2bp3* embryos might be not dependent on Ythdf2 and miR-430.

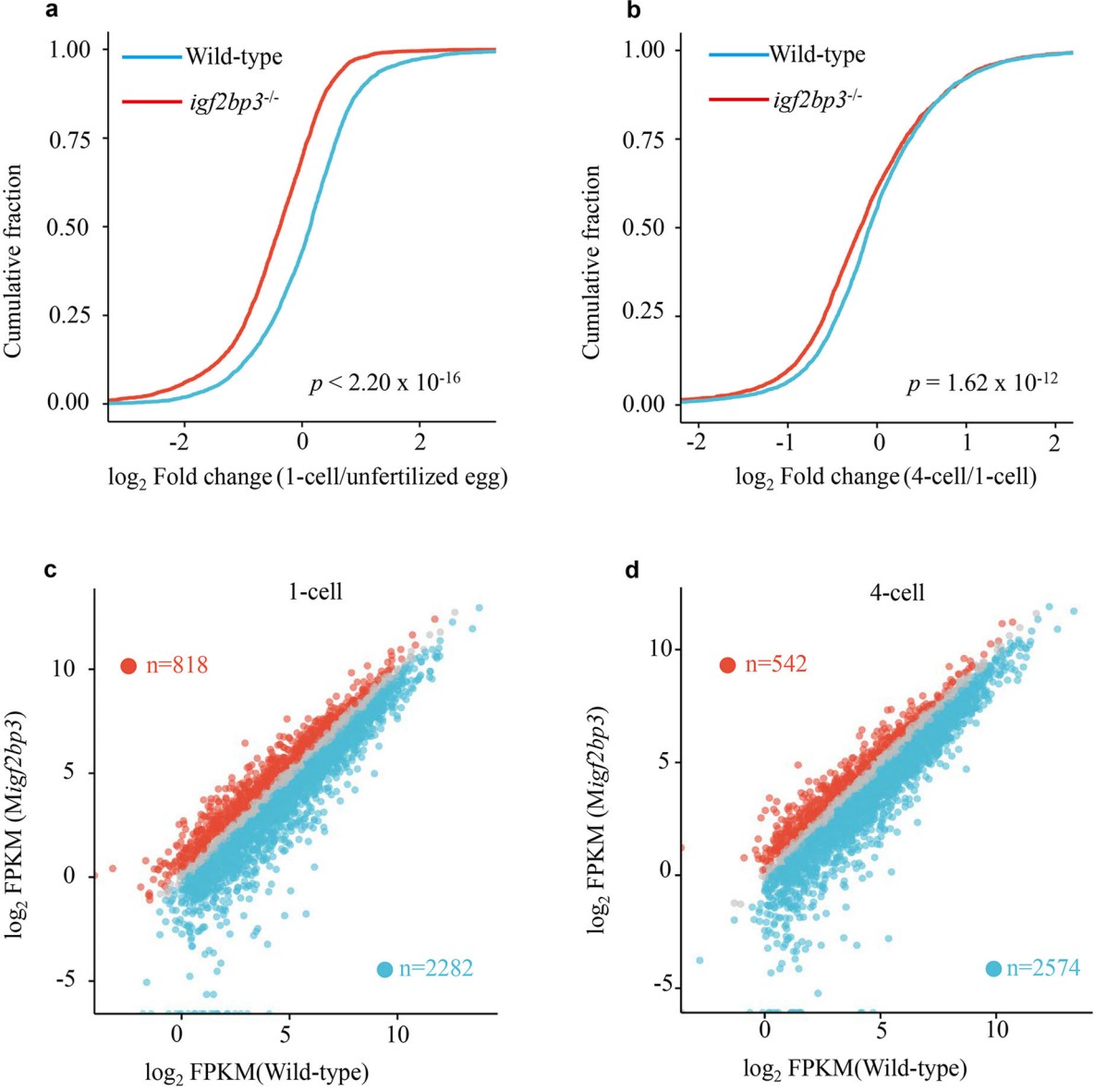

**Fig. 3 *igf2bp3* deficiency results in accelerated decay of maternal mRNA. a**, **b** Cumulative frequency of maternal mRNA log$_2$ fold changes in wild type and M*igf2bp3* from unfertilized eggs to 1-cell-stage embryos and 1- to 4-cell embryos, respectively. The *p* values were calculated using two-sided Wilcoxon and Mann–Whitney test. **c**, **d** Scatter plots showing the enrichment of dysregulated genes in wild type and M*igf2bp3* embryos at 1- and 4-cell stages. The numbers of significantly downregulated genes (blue dots) or upregulated genes (red dots) were shown.

***igf2bp3* overexpression decelerates the decay of maternal mRNAs**. To further determine whether Igf2bp3 regulates the stability of the maternal RNAs and early embryogenesis, we overexpressed *igf2bp3* mRNA in wild-type embryos. The embryos injected with *igf2bp3* mRNA exhibited a normal developmental process compared to the control (embryos injected with *GFP* mRNA) from fertilization to 64-cell stage. However, phenotype of developmental delay was detected at sphere stage and apparent at shield stage in *igf2bp3*-overexpressed embryos (Fig. 6a). The percentage of delayed embryos was calculated at shield stage and shown in Fig. 6b. After the injection of *igf2bp3-HA* mRNA into wild-type embryos, the protein started to be expressed at 4-cell stage (Supplementary Fig. 8). To investigate whether *igf2bp3* overexpression enhanced mRNA stability, we performed qPCR to detect RNA abundance of a set of maternal mRNA. The results showed that ectopic expression of *igf2bp3* decelerated the decay of these maternal mRNAs, instead of rapid clearance in the control

(Fig. 6c). These results were consistent with previous studies that embryos displayed developmental delay due to uncleared maternal RNAs[6], and suggested *igf2bp3* as an RNA stabilizer to regulate maternal RNA stability.

## Discussion
In vertebrates, maternal mRNAs accumulate in the egg and orchestrate subsequent embryonic development. Usually, the maternal mRNAs are stable for the first few hours of embryo development, and then are subjected to degradation during the MZT[3]. However, the mechanism of how maternal mRNAs keep stability pre-MZT is still unknown. In this study, we found that zebrafish Igf2bp3, an RBP, regulates zebrafish early embryogenesis through maintaining the stability of maternal mRNAs. The accelerated decay of maternal mRNAs and developmental defects occurred in M*igf2bp3* embryos at the pre-MZT phase, as wave of

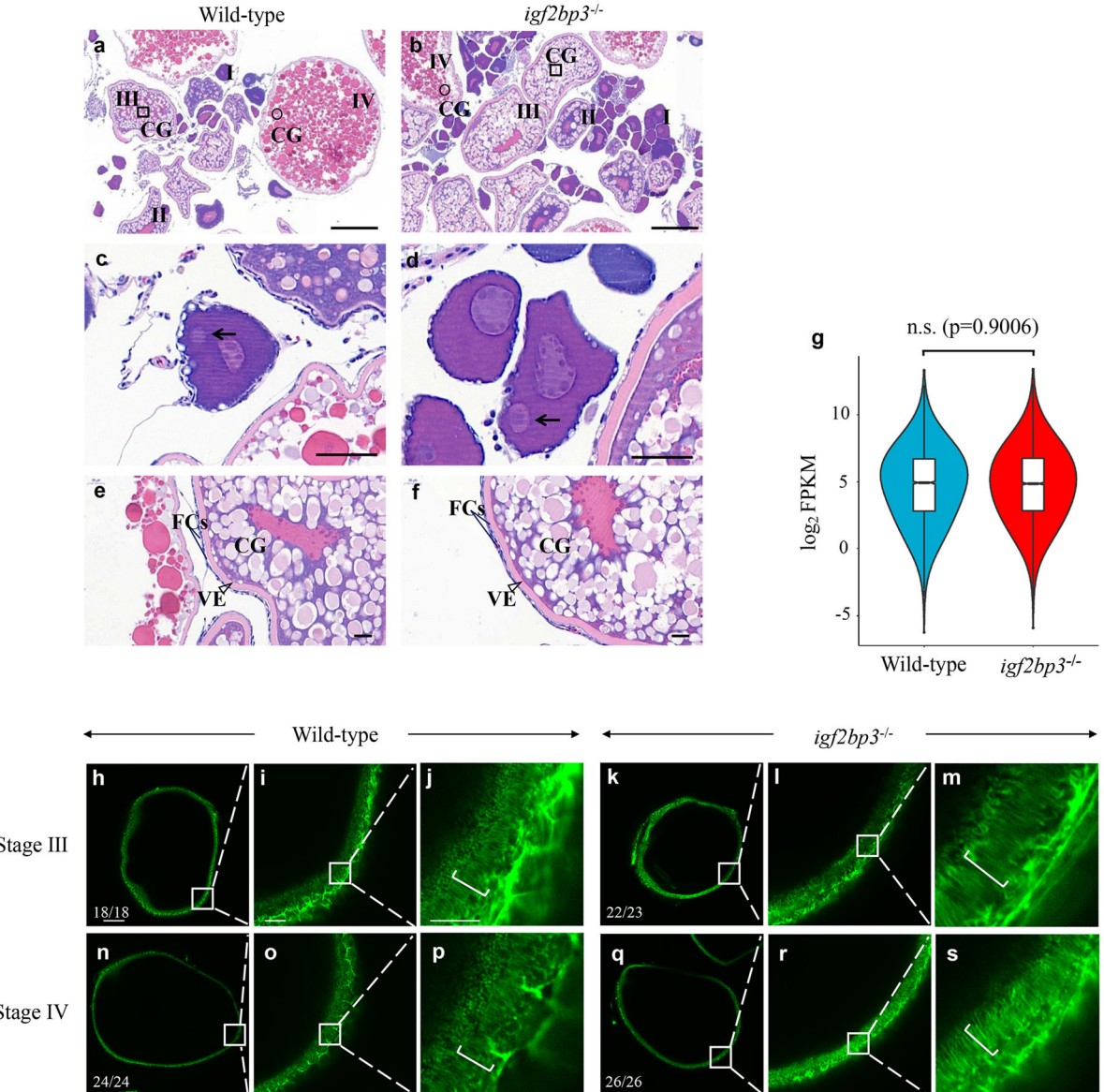

**Fig. 4 Oogenesis is largely unaffected in *igf2bp3* homozygous mutant (*igf2bp3*<sup>−/−</sup>) zebrafish. a, b** H&E-stained ovary sections indicated cortical granule movement toward the cortex, structure of the vitelline envelope, and somatic follicle cells surrounding stage I–IV oocytes of wild type and *igf2bp3*<sup>−/−</sup> zebrafish. CGs cortical granules. Square box indicates CGs located centrally in young oocytes and round box indicates CGs in the cortex at late stage; Scale bars:100 μm. **c, d** The normal polarization of stage Ib oocytes as indicated by the presence of the Balbiani body (black arrowheads) in *igf2bp3*<sup>−/−</sup> zebrafish compared with the wild type. Scale bars: 50 μm. **e, f** H&E-stained stage III oocytes revealed cortical granule movement toward the cortex in wild type and *igf2bp3*<sup>−/−</sup> zebrafish. VE vitelline envelope (triangle box); FCs follicle cells (two straight lines). Scale bars: 20 μm. **g** Violin and box plots depicting the expression levels of maternal genes in unfertilized eggs. Lower/middle/upper position in box plots indicated 25/50/75% quantile, respectively. The violin width shows the gene density. Two-sided Wilcoxon and Mann–Whitney test was used. n.s. no significant. **h–s** F-actin expression in wild type and *igf2bp3* mutant oocytes. The inset showed a zoom-in of the boxed region. **i, l, o, r** In the subcortical regions of stage III–IV oocytes, the actin filaments became clearly distributed in both wild type and *igf2bp3* mutant oocytes. **j, m, p, s** High magnification images of actin filaments in the oocyte cytoplasm. The actin columns were arranged from the inner to the outer across the cortical cytoplasm and thereby the column was arranged parallel to each other (brackets). Scale bars: 100 μm in **h, k, n, q**; 25 μm in **i, l, o, r**; 10 μm in **j, m, p, s**.

mRNA degradation was observed at as early as 1-cell and 4-cell stages (Fig. 3).

The fates of m⁶A-modified mRNA were controversial and determined by the m⁶A readers including YTHDF2 and IGF2BPs[11]. The m⁶A-modified mRNAs were subjected to decay when binding to the YTHDF2[6], whereas they were stable when binding to IGF2BPs. More than 80% of IGF2BP targets had at least one m⁶A peak in human cells and IGF2BP bound to m⁶A-modified RNAs through KH3–4 domain[13]. However, other studies have argued that IGF2BP did not bind directly to m⁶A, but instead showed that enhanced binding was a nonspecific consequence of m⁶A-induced mRNA unfolding[23]. Meanwhile, RBPs (Ythdf2 and Igf2bp3) binding led to RNA structural changes, which played a central role in connecting transcription, translation, and RNA degradation[23]. During early zebrafish embryogenesis, *ythdf2* facilitated maternal mRNA clearance during the MZT[6], while our studies indicated that Igf2bp3 maintained maternal RNA stability.

The RNA-binding protein IGF2BP has also been shown to regulate cell polarity and migration by modulating actin

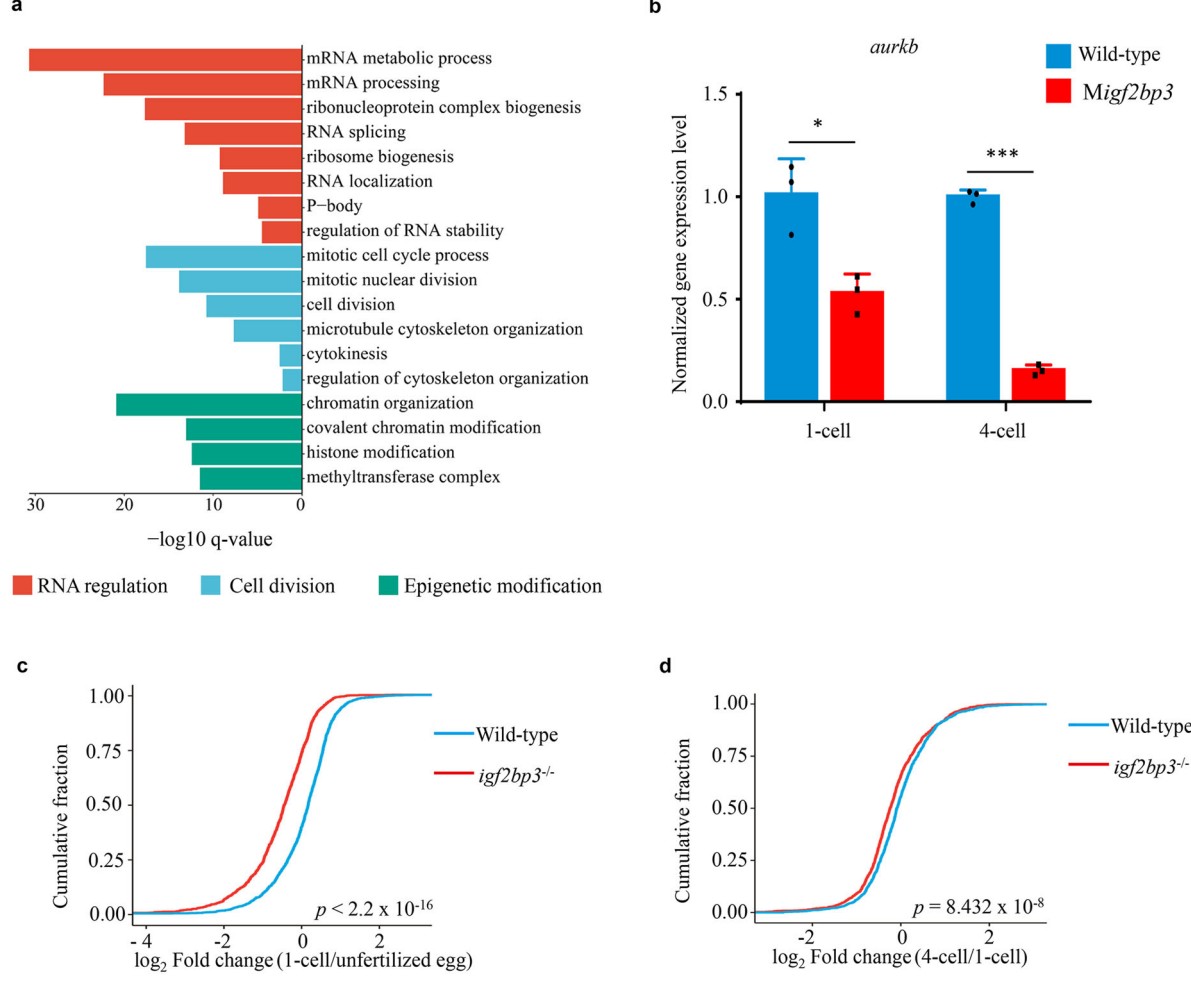

**Fig. 5 Igf2bp3 targets maternal mRNAs and regulates their stability in the early stage embryos. a** Gene Ontology analysis of the targets of Igf2bp3 in zebrafish embryo. **b** The expression levels of *aurkb* in the early development of zebrafish embryos. *P* values were calculated by two-sided Student's *t*-test. (**p* < 0.05, ****p* < 0.001). **c, d** The fold changes in expression levels of Igf2bp3 targeting maternal mRNA in wild type and M*igf2bp3* from unfertilized eggs to 1-cell-stage embryos and 1-cell to 4-cell embryos, respectively. *P* values were calculated by two-sided Wilcoxon and Mann–Whitney test.

cytoskeleton including β-actin mRNA translation and F-actin organization[24,25]. The *Drosophila* Igf2bp predominantly localized in the cytoplasmic RNP granules and bound to 3′ UTRs of many genes that were involved in F-actin formation. Knockdown of Igf2bp led to significant reduction of F-actin levels[26]. The F-actin and β-catenin protein expression in the cleavage furrow were severely disturbed in M*igf2bp3* embryos (Fig. 2a, b). The reorganization of F-actin cortex was associated with CG exocytosis[19,27]. By interacting with E-cadherin and the F-actin-binding protein αE-catenin during cellular cytoskeleton, β-catenin formed a cadherin–catenin complex to directly bind to F-actin[28]. Moreover, the target genes of Igf2bp3 were enriched in the process of cell division, cytoskeleton organization and cytokinesis (Fig. 5a). For example, the mRNA level of *aurkb* was significantly reduced in M*igf2bp3* embryos. However, hnRNP I, an important gene controls the process of CGE[29], was upregulated in the early stage of M*igf2bp3* embryos from the RNA-seq data, suggesting that *igf2bp3* regulated CGE in another unknown pathway. Accordingly, we observed severe developmental defects in both actin organization in cleavage furrow and CG release when loss of maternal *igf2bp3* function in the early zebrafish embryos.

Maternal mRNA degradation during MZT is a fundamental biological process that has been extensively studied in vertebrates[30,31]. Several RBPs, microRNAs, and transcriptional factors, such as Ythdf2, miR-430, and TBP are required for the degradation of maternal RNA[5,6,32], each representing distinct regulatory mechanism and controlling distinct sets of mRNAs. The waves of maternal mRNA degradation occurred at pre-MZT phase in M*igf2bp3* embryos (Fig. 4a, b), when the mechanism of maternal mRNA degradation is unknown. Interestingly, overexpression of *igf2bp3* led to delayed decay of maternal mRNAs and embryos development, which phenotype was similar to the phenotype of zebrafish mutants of *ythdf2*[6]. However, a large majority of the regulatory and target mRNAs of Igf2bp3, Ythdf2, and miR-430 were different (Supplementary Fig. 7), suggesting that the accelerated decay of maternal mRNAs in M*igf2bp3* embryos was in an alternative way, not dependent on the Ythdf2 or miR-430 pathway. In conclusion, our results indicate that RBP Igf2bp3 has a powerful function in maintaining the stability of maternal mRNA at pre-MZT stage. It remains to be elucidated the degradation mechanism of Igf2bp3-bound maternal mRNAs.

## Methods

**Zebrafish maintenance**. AB line wild-type zebrafish were maintained at 28.5 °C on a 14 h light/10 h dark cycle. All embryos used were collected by natural spawning and staged according to standard procedures[33].

**CRISPR/Cas9 technology and mRNA microinjection**. *igf2bp3* mutants were created using CRISPR/Cas9-mediated mutagenesis. The *igf2bp3* target site in the first exon was designed by the online service of the website (http://crispr.mit.edu).

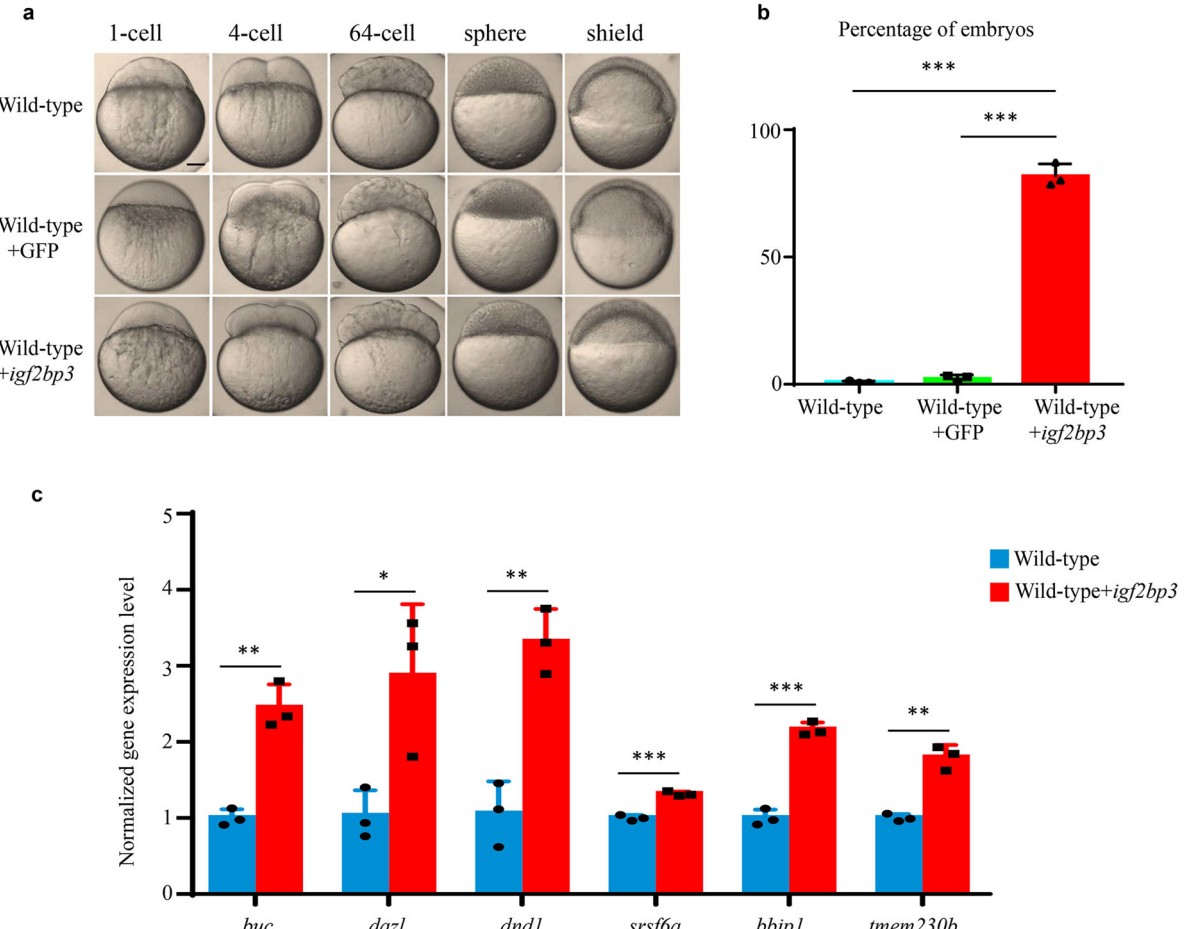

**Fig. 6 Overexpression of *igf2bp3* leads to enhanced maternal mRNA stability and causes developmental delay in wild-type embryos. a** Time-matched bright field images of embryos indicated that *igf2bp3*-overexpressing embryos experienced a developmental delay at early stages. Scale bars: 200 μm. **b** The percentage of embryos with developmental delay at shield stage. The embryos exhibited delayed development in non-injected groups (N1 = 3/572, N2 = 7/515, N3 = 19/622), in *GFP* mRNA injected groups (N1 = 12/411, N2 = 4/460, N3 = 16/487), and in *igf2bp3* mRNA injected groups (N1 = 378/481, N2 = 412/514, N3 = 439/503). **c** qRT-PCR analysis of maternal mRNA decay in wild type and *igf2bp3*-overexpressing embryos. mRNA abundance was normalized to values of wild type. Error bars represent mean±S.D., n = 3. *P* values were calculated by two-sided Student's *t*-test (\**p* < 0.05, \*\**p* < 0.01, \*\*\**p* < 0.001).

The DNA templates for gRNA transcription were amplified by PCR technology with synthesized primers gRNA-F: GTAATACGACTCACTATATG TGGATTGCCCCGACGAGAGTTTTAGAGCTAGAAATAGC and gRNA-R: AAAA GCACCGACTCGGTGCC. The purified PCR amplification was transcribed into gRNA using the transcript Aid T7 High Yield Transcript kit (Thermo Scientific, USA) as described[34]. Cas9 plasmid was linearized by *Xba*I enzyme, purified and transcribed into Cas9 mRNA using the T7 mMESSAGE mMACHINE Kit (Ambition, USA). Cas9 and gRNA mRNAs were purified by lithium chloride precipitation. Three hundred picograms Cas9 and 20 pg gRNA mRNA were co-inject into zebrafish embryos at 1-cell stage. The primer vF: AGTCGCATCGC CAAAGAGTA and vR: CCTTGCCCTTCAGTGGTTCT were used for genotyping. *igf2bp3-HA* was subcloned into the pCS2+ vector for in vitro transcription. Capped sense RNAs were synthesized with the mMESSAGE mMACHINE SP6 Transcription Kit (Ambion, AM1340). For overexpression experiments, approximately 1 nl of various concentration of *igf2bp3* or GFP mRNA were injected into wild type at 1-cell-stage embryos.

**Quantitative RT-PCR and whole-mount in situ hybridization**. Total RNA was isolated from each tissue and embryo sample using TRIzol reagent and was treated with DNaseI (Promega). For evaluating tissue expression of *igf2bp3*, Oligo-dT primer (Promega) was used to synthesize cDNA using GoScript™ Reverse Transcription System (Promega). To check the expression of maternal genes during embryo development, the RNAs were reverse transcribed into cDNA with PrimeScript RT reagent kit (Takara, RR047A) that uses primer mix containing both Random 6 mers and Oligo-dT Primer. qRT-PCR was performed using Light Cycler® 480 DNA SYBR Green I Master Mix (Roche) on a Light Cycler® 480 Instrument (Roche). *actb1* was used as the internal control in RNA degradation assay according to the previous report[6], and *18s* is used as the internal control in other procedures. Each experiment was performed in triplicate and the data were

analyzed using the $2^{-\Delta\Delta Ct}$ program. The gene-specific primer sequences are listed in Supplementary Table 1.

Embryos at different stages were collected and fixed as described previously[35]. Anti-sense probes were generated directly from PCR products containing a T7 RNA polymerase binding sequence at the 5′ end. Primer sequences are listed in Supplementary Table 1. Probe hybridization was performed as described previously[36]. Whole-mount embryos were imaged using Leica DFC550 with LAS 4.6 software.

**Immunostaining and immunofluorescence**. After fixed with 4% paraformaldehyde (PFA), F-actin of zebrafish embryos was detected with 0.22 μM Alexa Fluor 488 phalloidin (Cell Signaling, #8878), as previously described[4]. To stain F-actin in zebrafish ovary, dissected ovaries were fixed with actin stabilizing buffer and performed as described[37]. CGs were stained with 100 μg/ml TRITC-conjugated Maclura pomifera lectin (EY Labs, F-3901–2) for 30 min at RT, and washed with PBST for three times. For whole-mount immunofluorescence, anti-β-Catenin (Abcam, ab16051) and anti-α-Tubulin (Sigma, T5168) were diluted at 1:1000 and 1:2000, and then staining was performed according to the previous description[4,38]. The images of embryos were acquired by a Leica confocal microscope (TCS SP8) using LAS X software with 3D view model.

**Ovary histology**. Hematoxylin & eosin (H&E) staining was performed as previously described[39]. Ovaries were dissected from anesthetized adult females and fixed in 4% PFA overnight. Tissues were dehydrated and embedded in paraffin, sectioned (10 μm thickness), and stained with hematoxylin & eosin solution. Images were acquired using a 3D Pannoramic MIDI and an ECLIPSE Ci camera (Nikon).

**RIP and RIP-qPCR**. Wild-type embryos were injected with *igf2bp3-HA* mRNA at 1-cell stage and collected at sphere stage (more than 5000 embryos), when egg

membrane and yolk were removed by pronase E and buffer 1 (55 mM NaCl, 108 mM KCl, 1.25 mM NaHCO3). RIP was performed as previously described[40] with some modifications. The embryos were lysed in RIP buffer (50 mM Tris, pH 7.4, 150 mM NaCl, 5 mM $MgCl_2$, 0.5% NP40, 1 mM DTT with protease inhibitors Sigma P8340) with Murine RNase inhibitor (New England Biolabs, M0314). One microgram HA antibody (Cell Signaling, 3724) and IgG (as control, Millipore NI01) were incubated with Dynabeads Protein A (Thermo Fisher, 10001D) in lysis buffer and washed twice with lysis buffer. After centrifugation, the supernatant of embryo lysate was transferred to antibody-conjugated magnetic beads and rotated for incubation. After washing twice with lysis buffer and twice with wash buffer (50 mM Tris, 150 mM NaCl, 1 mM $MgCl_2$, 0.5% NP40), the beads were eluted in wash buffer containing 1% SDS and proteinase K. At last, the RNA was purified with RNA cleanup kit (Zymo Research, R1015) according to the protocol. The RNAs were analyzed by qRT-PCR or RNA-seq. The Igf2bp3-bound RNAs were reverse transcribed into cDNA using PrimeScript RT reagent kit (Takara, RR047A) and used for qRT-PCR analysis with the primers listed in Supplementary Table 1.

**Western blot**. After removing egg membranes and yolk of embryos with pronase E and buffer 1 (55 mM NaCl, 108 mM KCl, 1.25 mM $NaHCO_3$), 50–100 embryos were collected and lysed with 300 μL of cold lysis buffer (50 mM Tris at pH 7.5, 150 mM NaCl, 1 mM EDTA, 10% glycerol, 1% Triton X-100 and protease inhibitors, Sigma P8340). An equal amount protein of each sample was separated in 10% SDS-PAGE gel, transferred onto PVDF membranes, and blotted with anti-HA antibody (Cell Signaling, 3724) and anti-β-actin antibody (ABclonal, AC026). The blots were detected with HRP-conjugated anti-rabbit IgG (H + L) and HRP-conjugated anti-mouse IgG (H + L) and visualized using ECL Substrate (BIO-RAD, 170-5061).

**RNA-seq and data processing of high-throughput sequencing**. The total RNA of unfertilized eggs and embryos at different developmental stages of wild type and mutant zebrafish were isolated with TRIzol reagent, and mRNA was purified from total RNA as described previously[41]. Purified mRNAs and RIP-RNAs were fragmented and RNA library was constructed using NEBNext® Ultra II Directional RNA Library Prep Kit for Illumina (NEB, E7760) according to the manufacturer's instructions. RNA-seq and RIP-seq were sequenced on Illumina NovaSeq 6000 or HiSeq X-ten platform with paired-end 150 bp read length.

For RNA-seq analysis, raw paired-end reads were filtered using fastp 0.19.7 under default parameters[42]. The filtering steps included the removal of reading pairs if either one read contained adapter contamination. Filtered reads were mapped to zebrafish genome version 11 (GRCz11) using STAR 2.7.2b[43] with command line options "--twopassMode Basic --outFilterMultimapNmax 10 --quantMode TranscriptomeSAM GeneCounts". The gene-level quantification was performed using RSEM 1.3.1[44] with the default parameters for paired-end RNA-seq. To identify maternal and zygotic genes, Cluster3.0 (ref. [45]) was used to divide all expressed genes into three clusters according to the previous study[6]. miR-430- and Ythdf2-mediated degradation profile was analyzed from the published RNA-seq data[5,6].

For RIP-Seq analysis, raw reads were adapter trimmed with cutadapt[46] using the additional parameters "-e 0.1 -m 10 --quality-cutoff 6" and subsequently mapped to zebrafish GRCz11 reference genome using STAR 2.7.2b with command line options "--outFilterMultimapNmax 30 --outFilterMultimapScoreRange 1 --outFilterType BySJout --outFilterScoreMin 10 --alignEndsType EndToEnd". Mapped reads were assigned to zebrafish ensemble genes using featureCounts[47]. The edgeR Bioconductor package[48] in the R statistical computing environment[49] was used for calculating FPKM and differential expression analysis. We defined the Igf2bp3 targets as those with FPKM > 1 in IP samples (HA-IP or IgG-IP) and were upregulated >2-fold in HA-IP sample compared with IgG-IP sample. Gene Ontology enrichment analysis of Igf2bp3 targets was carried out using clusterProfiler[50], significant GO terms were defined as having Benjamini–Hochberg[51] adjusted p value below 0.05. A summary of sequenced samples and processed data are included in Supplementary Data 1.

**Statistics and reproducibility**. All experiments were performed by independent replicates and the results of replicates were consistent. Sample sizes and groups were present in figure legends and source data. Statistical analysis was carried out using R and GraphPad Prism 7. All data were analyzed using the two-sided Wilcoxon and Mann–Whitney test or unpaired Student's t-test. Error bars represent standard deviation. P value of <0.05 was considered as significant (*), while P < 0.01 and P < 0.001 as extremely significant (**) and (***), respectively.

**Reporting summary**. Further information on research design is available in the Nature Research Reporting Summary linked to this article.

## Data availability
RNA-seq data and RIP-Seq data reported in this article could be accessed at NCBI BioProject under accession number PRJNA565584. Additionally, all data used for figures were provided as source data in Supplementary Data 2.

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

## Acknowledgements

This work was supported by the Fundamental Research Funds for the Central Universities [534180010235, 2662017PY013], the National Key R&D Program of China (No. 2018YFD0901201), and China Agricultural Research System (CARS-46).

## Author contributions

J.M., J.G., and F.R. designed the overall study, F.R., Q.L., G.G., X.D., H.D., W.Q., R.M., Y.X., X.L., and R.X. performed experiments, collected and analyzed data. J.M. and F.R. wrote the paper with input from all the other authors.

## Competing interests

The authors declare no competing interests.
