## [Peer Review File · Communications Biology]

Editorial Note: *This manuscript has been previously reviewed at another Nature Research journal. This document only contains reviewer comments and rebuttal letters for versions considered at Communications Biology.*

REVIEWERS' COMMENTS:

Reviewer #1 (Remarks to the Author):

Overall, I think this manuscript is improved by removing the link to m6A which was very weak and not convincing. The results here are more descriptive, but nevertheless useful in understanding the role of Igf2BP3 in early embryo development in zebrafish.

I would only make one suggestion, which is textual in nature. The authors cite reference 13 in line 267, page 10, to indicate that IGF2BP binds to m6A-modified RNA through the KH domain. However, as mentioned before, newer studies have shown that this is an artifact. It is important to be balanced and present both the views, even if it is a discussion point that is not central to your manuscript. You can simply say, "however, other studies have argued that IGF2BP does not bind directly to m6A at all, but instead shows enhanced binding is a nonspecific consequence of m6A-induced mRNA unfolding." There, you can cite the paper by Howard Chang and Q. Zheng. This way the readers won't be misled and the authors will not be perpetuating a possibly incorrect finding. A statement like this allows the authors to be fair to everybody.

Reviewer #1 (Remarks to the Author):

Overall, I think this manuscript is improved by removing the link to m6A which was very weak and not convincing. The results here are more descriptive, but nevertheless useful in understanding the role of Igf2BP3 in early embryo development in zebrafish.

I would only make one suggestion, which is textual in nature. The authors cite reference 13 in line 267, page 10, to indicate that IGF2BP binds to m6A-modified RNA through the KH domain. However, as mentioned before, newer studies have shown that this is an artifact. It is important to be balanced and present both the views, even if it is a discussion point that is not central to your manuscript. You can simply say, "however, other studies have argued that IGF2BP does not bind directly to m6A at all, but instead shows enhance binding is a nonspecific consequence of m6A-induced mRNA unfolding." There, you can cite the paper by Howard Chang and Q. Zheng. This way the readers won't be misled and the authors will not be perpetuating a possibly incorrect finding. A statement like this allows the authors to be fair to everybody.

Thanks for the suggestion. We have cited the paper by Howard Chang and Q Zheng and modified the discussion.